# The Effects of Virtual Reality on Enhancement of Self-Compassion and Self-Protection, and Reduction of Self-Criticism: A Systematic Review

**DOI:** 10.3390/ijerph20032592

**Published:** 2023-01-31

**Authors:** Tomáš Žilinský, Júlia Halamová

**Affiliations:** Institute of Applied Psychology, Faculty of Social and Economic Sciences, Comenius University in Bratislava, Mlynské luhy 4, 821 05 Bratislava, Slovakia

**Keywords:** self-compassion, self-criticism, self-protection, systematic review, virtual reality

## Abstract

Background: Virtual reality used for the treatment of mental health disorders is showing promising potential in clinical practice. Increasing self-compassion and self-protections and decreasing self-criticism have been identified as trans-diagnostic mechanisms helping to build a resilient self. The goal of this systematic review was to provide an overview of research studies about virtual reality while exploring its effectiveness in increasing self-compassion and self-protection and decreasing self-criticism. Methods: On 6 December 2022, titles, abstracts, and, where available, keywords were searched in the following databases: PubMed, Scopus, and Web of Science. The inclusion criteria were: empirical study; quantitative methodology; outcomes measured, including self-compassion and/or self-protection, and/or self-criticism; pre/post and/or control group intervention measures of selected outcomes; participants aged 18 and above; application of virtual reality technology as part of the intervention; full study availability; and English language. Exclusion criteria were: ruminations related only to physical pain; self-protection in the context of physical survival; VR used to increase ruminations. Results: Selection criteria were met by 19 studies (two randomized controlled trials, 16 longitudinal studies, and one case study) with an overall number of 672 participants. Results suggest that VR interventions targeting self-criticism, self-compassion, and self-protection might be effective for non-clinical (self-compassion) as well as clinical (self-criticism and self-protection) samples. Discussion: The present systematic review partially supports the effectiveness of VR interventions on self-criticism, self-protection, and self-compassion. To properly answer the question of its effectiveness, more randomized control trials with larger samples from different populations are needed. The results are also limited by the variability of intervention protocols and the amount of exposure to VR. Other: This work was supported by the Vedecká grantová agentúra VEGA under Grant 1/0075/19. This systematic review has not been pre-registered.

## 1. Introduction

Virtual reality (VR), as a facilitator of psychological interventions for the treatment of mental health disorders, is increasingly showing potential for clinical practice [1,2]. This has been particularly the case in the last few years, as it is only since 2016 that high-quality immersive VR technology has become affordable, making it accessible to wider audiences [3].

VR is defined as a technology that enables an advanced interaction between humans and computers and allows for humans to become immersed in a synthetic environment generated by the computer [4]. Although VR is meant to be an interactive and immersive system by design, the various existing forms of VR on the market can be divided into immersive and non-immersive, as well as interactive and non-interactive VR. Non-immersive VR is usually delivered through a computer screen and tends to come in the form of a three-dimensional computer game controlled via a keyboard, joystick, or some other type of interface [5]. Immersive VR, alternatively, can be described as a life-sized environment that evokes in the user the sensation of actually being part of it [6]. Immersion is usually achieved by blocking out sensory stimuli from the physical world and presenting an alternative three-dimensional digital world through a head-mounted display (HMD) or through an immersive room, such as Cave Automatic Virtual Environment (CAVE) [7]. Interactive VR is a system that is essentially sensitive and responsive to the user’s behavior, facilitating an interaction between the user and objects within this virtual world. Conversely, non-interactive VR does not allow for interaction; the user is more-or-less a passive viewer of a digitally generated environment. A 360-degree video would be considered non-interactive VR: although the user can exploit the full view through movements (e.g., head rotation while wearing a HMD), he or she cannot interact with the content of the presented environment.

VR has the capacity to provide a subjective and illusional experience, i.e., to (to some degree) simulate reality and real-world interactions, which indicates broad scientific and clinical implications. Moreover, VR offers tight control over simulation, reduces the inconsistency of interventions, and, most importantly, allows for the design of tailor-made therapeutically valuable situations that are otherwise (nearly) impossible to recreate in real life [6]. So far, research has been conducted on the clinical use of VR in the treatment of mental health disorders, showing promising results for depression (e.g., [8]), anxieties (e.g., [9]), post-traumatic stress disorder (e.g., [10]), eating disorders (e.g., [11]), pain management (e.g., [12]), autism (e.g., [13]), forensic psychiatry (e.g., [14]), and schizophrenia (e.g., [15]).

Depression and anxiety are two of the most common mental health disorders globally [16]. They frequently coexist—depressive people often have symptoms of anxiety, just as people with anxiety disorders often suffer from depression [17]. One of the main risk factors predicting both depression and anxiety has been identified as self-criticism [18]. Self-criticism can be conceptualized as an intense and persistent relationship one has with oneself, consisting of an unshakeable and increasing demand to constantly perform within the highest standards and an inwardly-directed hostility and depreciation when this unachievable demand is not met [19]. Self-criticism is often interchangeably used with or related to negative ruminations, self-judgement, self-hatred, self-contempt, self-attack, the inner critic, or self-shame. As the antidote to the pain that self-criticism (or negative self-treatment) causes, two adaptive processes were identified—self-compassion and self-protection—both within the framework of Emotion-focused Therapy [20]. In the therapeutic model of Emotion-focused Therapy [21], self-criticism, self-compassion, and self-protection are meaningfully tied together, and working with them is central to building one’s empowered and resilient self [20]. Emotion-focused Therapy’s efficiency for the treatment of self-criticism was supported, for example, by Shahar et al. [22,23], Thompson and Girz [24], and Timulak et al. [25].

Self-compassion is best understood as inwardly-directed compassion [26]. It can be conceptualized as an emotion [27,28], frame of mind [29], skills set [30], or process [31] that is associated with understanding the universality of human suffering, recognizing one’s own suffering, feeling for the suffering person, tolerating uncomfortable feelings, and acting on or having the motivation to act on that suffering in order to alleviate it. Interchangeably with self-compassion, scholars also use terms such as self-soothing or self-reassurance. Self-protection can be understood as assertiveness directed at one’s own inner critic. Also labelled as assertive anger [32] or protective anger [20], self-protection can be defined as a state or adaptive emotion that allows a person to stand up for their rights and needs, set boundaries against, and assertively engage in a fight with their own mistreatment they have internalized from the voices of important figures, mainly in childhood. More recently, Neff [33] introduced a similar concept of fierce self-compassion, which is defined as a united caring force against environmental pressures and injustices but also against one’s own habits. Interest in self-compassion has been growing significantly since the early 2000s [34], with the first few studies employing VR to enhance self-compassion appearing more recently (e.g., [35]). So far, it seems that a lot less has been published about self-protection in general. However, assertiveness most likely entered VR research at the turn of the millennium [11].

This systematic review focuses on VR interventions targeting the three abovementioned self-concepts: self-criticism, self-compassion, and self-protection. This systematic review summarizes the existing scientific literature on VR interventions applied to decrease self-criticism or increase self-compassion or self-protection, compared with control treatment, no treatment, or baseline scores in the clinical and non-clinical adult populations. The results of this systematic review may inform the development of effective VR interventions targeting not only these three self-concepts but ultimately may lead to new ways of addressing depression and anxiety. To our knowledge, this is the first systematic review focusing on self-compassion, self-criticism, and self-protection applied in virtual reality. As results of using virtual reality for mental health interventions are very promising, and these three concepts play a pivotal role in psychopathology as well as mental and physical health, we were particularly interested in how effective VR is for these particular three concepts.

## 2. Methods

This systematic review was directed according to the Preferred Reporting Items for Systematic Review and Meta-Analyses (PRISMA), using the 2020 updated guidelines [36].

### 2.1. Information Sources

This systematic review includes all literature written in English, published until 6th December 2022, and accessible through the PubMed, Scopus, and Web of Science databases. The main focus was on VR as a platform facilitating the increase of self-compassion and/or self-protection and/or the decrease of self-criticism. In all three databases, records (titles and abstracts) were searched, and where available, keywords were added to the search too.

### 2.2. Search Strategy

The searches were completed in three separate steps, depending on the search topic. The general search terms for all three searches were: (“virtual reality” OR “VR” OR “virtual environment*” OR “virtual world*” OR “avatar” OR “serious game*”). The topic-specific search terms for self-compassion were: (“compassion*” OR “self-compassion*” OR “self-sooth*” OR “self-reassur*”); for self-criticism: (“self-critic*” OR “inner critic*” OR “self-contempt*” OR “ “self-hat*” OR “self-attack*” OR “ruminat*” OR “self-judg*”); and for self-protection: (“self-protecti*” OR “protective anger” OR “assertive anger” OR “assertive*” OR “self-assert*” OR “fierce self-compassion”). See Appendix A for the full search syntax used in each of the databases.

### 2.3. Eligibility Criteria

The inclusion criteria were: empirical study; quantitative methodology; outcomes measured, including self-compassion and/or self-protection/assertiveness, and/or self-criticism/negative ruminations; pre/post and/or control group intervention measures of selected outcomes; participants aged 18 and above; application of VR technology as part of the intervention; full study availability; and English language. Exclusion criteria were: ruminations related only to physical pain; self-protection in the context of physical survival; and VR used to increase ruminations. When synthesizing the evidence, studies were grouped according to the main outcomes of this review: self-criticism, self-compassion, and self-protection.

### 2.4. Selection Process

The eligibility criteria were decided on by both authors. The database search was conducted by the first author (TŽ). The screening was completed by the first author under full supervision of the second author (JH). The first author justified all selected and rejected studies to the second author. The second author challenged the first author’s justifications. All decisions on inclusion and exclusion were made consensually by both authors.

### 2.5. Data Collection

All variables were agreed on by both authors. The data collection process was conducted by the first author and checked by the second author. The first author consulted the second author about any uncertainties or inconsistencies that arose from the data collection process. Only variables relevant to this systematic review were extracted. If other measurements had taken place in the selected studies, these were not included. In total, nine variables had been decided: one outcome domain (#8) and 8 additional variables (#1-#7, #9):Type of the study design; studies were defined according to the purpose of this systematic review (i.e., even though a study had a control group, if the control group was not used as a comparison for the intervention group in respect to our outcomes, we treated it as a within-group study).The components of each intervention and, where available, the duration of each component; and the total duration of VR exposure one participant in the intervention group received.Whether participants in the control group received VR exposure too or not.The content of each VR intervention.Immersion and interactivity of VR technology used.The type of sample used in the study and their age.The size of intervention and control group(s).What outcomes, according to the focus of this systematic review, were measured (self-compassion, self-protection, assertiveness, negative ruminations, or self-criticism) and how (names of relevant measuring tools used, if available).The results of relevant outcomes as measured in point #8.

Not all studies reported all variables in full detail. Where available, study investigators were contacted directly via email for any missing information. We define only immersive/non-immersive and interactive/non-interactive VR technology as, likely, this is the most straightforward way to understand VR technology from a non-technical user experience point of view. There are, of course, other parameters (e.g., display resolution, manufacturer, etc.) that could be looked at when classifying VR technology; however, this would perhaps make a lot more sense once there are more studies published.

### 2.6. Risk of Bias and Certainty Assessment

The risk of bias was assessed by the first author and checked and challenged by the second author. Discrepancies were discussed, clarified, and agreed upon together. As a framework, we followed the Cochrane Collaboration’s tool for assessing risk of bias in interventions [37], looking at selection, performance, detection, attrition, reporting, and other biases. We used this tool for both randomized and non-randomized studies. To assess attrition, we followed the five-and-20 rule of thumb used by some researchers, considering attrition of less than 5% as posing little threat and attrition greater than 20% as posing a potentially serious threat to the results’ validity [38].

Additionally, the certainty assessment for all outcomes was assessed by the first author and checked and challenged by the second author. Discrepancies were discussed, clarified, and agreed upon together. Certainty assessment was completed using the Grading of Recommendations Assessment, Development, and Evaluation (GRADE) framework [39], used for rating the quality of evidence in systematic reviews. The risk of bias due to missing results (publication bias) was assessed as part of the GRADE framework [39].

## 3. Results

A total of 468 records were identified through our database searches. The exact search breakdown can be seen in Table 1. Selection criteria were met by 19 articles, of which two articles related to seemingly identical research [40,41]; hence, the final selection list is composed of 18 articles. However, one study [42] included different measurements of relevant variables at different time points: one measurement with one particular psychometric tool (MSCS; [43]) was taken at time points T1 and T4, and a different measurement using a different psychometric tool (SOFI; [44]) was taken at time points T2 and T3. In this case, we would treat them as two separate studies, making the final total of selected studies 19. See Table 2 for the complete list of studies in alphabetical order and the structured summaries for each variable.

The most common reasons for reports not to be selected for this review were: selected variables not measured; no empirical study; no human subjects involved; VR not used; participants below the age of 18; and no pre/post and/or control group intervention measures of selected outcomes. For details, see the PRISMA flow diagram in Figure 1.

In terms of study designs, we looked at the studies from the perspective of our outcomes (self-compassion, self-protection, and self-criticism). If a study had a control group, but this control group was irrelevant to the outcome of our interest, we would treat it as a within-group study rather than a between-group study. Of the 19 selected studies, two were randomized controlled trials, 16 were longitudinal studies, and one was a case study. Of the 16 longitudinal studies, seven were randomized between-group studies, two were non-randomized between-group studies, one was a between-group study with randomization not specified, and six were within-group studies (see Table 2 for details). 

Overall, in terms of bias, each of the included relevant studies shows between none and two low risks of bias, three to five high risks of bias, and one to three unclear risks of bias out of the six predefined categories of bias [37]. For self-criticism, three to five high risks of bias and one to three unclear risks of bias were identified amongst the respective studies. For the self-compassion domain, three to five high risks of bias and one to three unclear risks of bias were identified among the respective studies. Lastly, for the self-protection domain, three to four high risks of bias and one to two unclear risks of bias were identified amongst the respective studies. This indicates that some concern is certainly needed when interpreting results across each of the domains of this review (see Appendix B for details and Table 3 for a summary). In terms of certainty assessment, all three domains were rated as very low, mainly due to the dominance of small studies that have been conducted to date and the large variability amongst them (see Table 4 for details).

Across all the studies, the overall number of participants taking part was 672, with 351 in the intervention and 321 in the control groups (these counts are only estimations as two studies did not provide the exact ratio between the intervention and the control group; [11,55]). Since some participants took part in two studies [42], we included them twice in our total count. Three studies [47,48,53] had a control group that was not relevant for measuring our outcomes; these participants were not included in the abovementioned counts. If a study had more than two groups, the group that received VR intervention was considered the intervention group, and the remaining groups were considered the control group. See Table 2 for details.

The final synthesis was grouped around each of the outcome domains: self-criticism, self-compassion, and self-protection. If a study measured two outcomes, e.g., self-criticism and self-compassion, the respective variables (as shown in Table 2) were synthesized for each outcome separately.

### 3.1. Self-Criticism

This systematic review identified six studies within the self-criticism outcome domain, with three of them measuring the impact of VR on self-criticism [8,35,50] and three on ruminations [47,48,54]. Of the six studies, four were published from 2016 onwards. Chronologically, the first study for this outcome looked at the application of VR to post-event processing in people with social anxiety disorder [54]. All six studies showed a decrease in self-criticism post-intervention; in five cases [8,35,47,48,54], this decrease was statistically significant. Additionally, four studies reported large effect sizes for self-criticism and ruminations decrease [8,35,47,50]; one study reported a large effect for ruminations decrease as part of a larger treatment group [54]; and one study did not report an effect size for ruminations at all [48].

All six studies used immersive head-mounted displays and interactive scenarios, and VR comprised the central part of all interventions. Four studies [8,35,47,48] used psychoeducation as part of the intervention, and three studies [8,35,50] applied embodied experience. On average, they had 24 (between 15 and 32) participants in the intervention groups. Four studies used clinical populations; in the remaining two cases [35,50], the populations were non-clinical. Three studies [8,47,48] did not have a control group, and one study [35] applied VR in the control group as well. The data on the number of completed VR sessions and overall durations spent in VR is not complete; from those reported, participants received between one and eight VR sessions with an overall VR exposure of 9.5 to 90 min. The scenarios involved either comforting a virtual child, embodying a patient with a severe anxiety disorder, or introducing oneself to or speaking to a virtual audience varying in size and level of disruption.

Results suggest that using immersive and interactive VR technology could be effective in targeting self-criticism. Additionally, psychoeducation may possibly play its part too as it was part of the intervention in four out of six studies; however, there is not enough evidence to either support or refute this observation. All studies with clinical samples showed significant results. All studies targeting the ruminations of participants suffering from social anxiety disorder (through virtual anxiety-provoking situations) reported significant reductions in ruminations. Where reported, studies with significant results delivered a minimum of three VR sessions with an overall VR exposure per participant of 24 min.

### 3.2. Self-Compassion

In terms of the outcome of self-compassion, nine studies were identified; eight of them measured self-compassion [8,35,42,45,49,50,51,53]; and one measured positive qualities towards self as a measure of self-compassion [42]. All studies, except for one [35], were published from 2016 onwards. Chronologically, the first study in this outcome domain applied VR embodiment experience to decrease self-criticism through self-compassion in women high in self-criticism [35]. Overall, of the nine studies, seven showed an increase in self-compassion [8,35,42,45,49,53]; for five of them, this increase was significant [8,35,45,49,53]. Of the remaining two studies, one showed a non-significant decrease in self-compassion [50] and one showed a non-significant change in self-compassion without reporting the direction of the effect [51]. Five studies [8,35,42,49] reported large effect sizes for self-compassion; one study [50] reported no effect for self-compassion; and three studies [45,51,53] did not report any effect size.

Of the nine studies in this domain, six [8,35,42,45,50] used immersive and interactive technology; two [49,51] used immersive head-mounted displays with non-interactive scenarios; and one [53] employed a non-immersive laptop with interactive content. In two studies, however, VR did not constitute a central part of the intervention [42,53]; in fact, it could be said that VR was marginal compared with the much larger proportion of other intervention parts. Embodied experience was part of the VR intervention in six studies [8,35,42,45,50]. Participants attended between one and six sessions, with all but three studies comprising only one VR session. Where reported, participants engaged in a minimum of 4 to 46 min of VR experience. The samples were, except for two studies [8,51], non-clinical, with an average intervention group size of 19 (between eight and 70). Three studies [8,51,53] had no control group, and two studies [35,45] had their control group also exposed to VR. Scenarios varied from an astronaut’s mission to the moon, observing people, avatars, objects, or nature, to embodying a child or an adult, healthy or with a mental condition.

Results suggest that VR could be effective in increasing self-compassion; looking at studies with significant results, this could be true even after just one 10 min, potentially even shorter, VR session. However, as all of the samples except for two were non-clinical, this can be concluded only for non-clinical adult samples. Technology, scenarios, and the structure of interventions varied across the studies.

### 3.3. Self-Protection

For this outcome domain, seven studies were identified; all of them measured the impact of VR on assertiveness [11,15,40,41,52,55,56,57]. All the studies were published before 2016. Chronologically, the first study in this outcome domain applied VR to treat body image disturbances in overweight women [11]. All seven studies showed a positive impact of VR on assertiveness; five studies [11,15,40,41,52,55] showed significant improvements; one study [57] fully reported only one measure, and it showed significant improvement for assertive behavior; and one study [56] did not report any inferential statistics. In terms of effect sizes, one study [57] reported a large effect for one of their assertiveness measures but did not report an effect for their second assertiveness measure; one study [40,41] reported a small and medium interaction effect between VR and CBT groups for their two assertiveness measures; another study [52] reported a large effect of time and a medium interaction effect between their VR and non-VR training groups; the remaining four studies did not report any effect sizes.

All seven studies employed interactive scenarios; however, only three of them [11,52,55] used head-mounted displays; the rest used personal computers, although in two cases [15,57] together with 3D glasses and headphones. In all studies, VR was a central part of the intervention. All participants attended between seven and 16 sessions, with each receiving approximately 220 to 480 min of VR experience in total; in one study [52] this information was not provided. VR was used only in intervention groups, which consisted solely of clinical samples averaging 13 participants (between one and 32). All studies but one [15] had a control group, and one study [57] was a case study of one patient. Scenarios included daily social situations and exposure to challenging stimuli.

Results suggest that VR could be effective in increasing self-protection/assertiveness, specifically when using VR technology with interactive scenarios that elicit challenging emotions in different virtual situations. However, based on this systematic review, this can be concluded only for clinical populations. Where reported, studies with significant results show a minimum of seven VR sessions with an overall VR exposure per participant of 220 min.

### 3.4. Adverse Effects

In addition to the analysis above, we also screened the studies in this review for any reports of adverse side effects as a result of taking part in VR, also known as cybersickness [70]. One study [51] reported adverse side effects (nausea and dizziness) in two participants; however, not necessarily attributable to VR intervention. Three studies [11,52,55] reported no cybersickness in any of their participants. The remaining studies did not address VR-related adverse side effects in their studies.

## 4. Discussion

This systematic review looked at the effectiveness of VR in decreasing self-criticism and increasing self-compassion and self-protection. Positive results have been identified across all three domains, adding to the encouragement of the application of VR for mental health purposes [5,6]. However, there are apparent differences (study design, number and duration of VR exposure, technology, scenarios, etc.) in the reviewed studies, and careful considerations need to be taken into account when interpreting these findings and offering any conclusions.

For decreasing self-criticism, immersive VR technology with interactive scenarios shows positive results after a relatively short-term intervention, specifically for adult clinical samples. This, or any other broader judgment, is however very limited, considering there were only six studies identified for this domain. It appears that this area has so far not received enough attention, considering the first study appeared in 2011. In addition, if there is a significant impact on clinical samples, it would be worth testing efficiency in non-clinical samples as well because the impact could be potentially even higher, as clinical samples may struggle more to overcome self-critical thoughts [71].

A brief VR intervention could be effective for increasing self-compassion in adult non-clinical samples. This conclusion comes from reviewing nine studies. These studies, however, varied greatly, indicating that more research is needed to better understand which features of VR intervention make it effective. An area to closely look at might be embodied experience, as already pointed out by Falconer et al. [35]. It seems promising that some of the studies with only one VR exposure made significant changes to self-compassion. If further elaborated, there is a big potential for improving the well-being of a greater non-clinical population. In addition, we suggest testing its potential with clinical samples as well.

For increasing self-protection/assertiveness, VR with interactive scenarios seems to have a positive effect, specifically in adult clinical samples. This, or any other conclusion, is again very limited, considering there were only seven studies identified for this domain. What stands out for this domain is that more repetitions and longer overall VR exposure took place compared with the other two domains. However, it is unclear whether fewer repetitions and a shorter overall exposure would have a different effect. With only seven studies to date, this area of research seems to have received very little attention since its onset in 2001, with the latest study to date published in 2014. It would be worth testing the impact of just one, or a shorter, VR exposure to develop more cost-effective interactions for increasing self-protection.

Interventions varied greatly across all the studies, and it remains unclear what is/are the key component(s) of effective VR interventions. Looking at the technology employed, the majority of the studies used immersive HMD, with only a few using a non-immersive computer screen. As none of the non-immersive studies provide information on the size of the effect, it is hard to judge what impact computer immersion plays on intervention effectiveness. Similarly, little can be concluded about interactivity, as only two studies provided non-interactive, passive content; the vast majority of the studies had their participants actively involved in the process of intervention. Additionally, the actual VR scenarios varied considerably in terms of content, number of exposures (repetition), and time spent in VR. This follows the observation of Rizzo and Koenig [5] that the identification of active ingredients in the use of clinical VR is still very much needed.

Another area of concern is the sample size. Small sample sizes can make statistical significance and effect size results unreliable. Field et al. [72] suggest a sample size of approximately 30 as a threshold for a reliable real-world population estimate. In our review, in all three domains, the average sample size was below this number, making any generalizations difficult. Additionally, more than a third of the studies were conducted on students, further adding to this issue. Larger sample studies with various population types are needed.

Furthermore, a great variety of self-report measures have been used across the selected studies, making a reliable synthesis difficult. Additionally, many of the selected studies show an overreliance on self-reported measures. Self-rated scales inevitably produce errors [72] and are confounded by bias, as reported by some of the authors in this review too (e.g., [48]). Therefore, the interpretation of results needs to take this point into account. Wider employment of other means of measurement (e.g., behavioral, vocal, or physiological markers) could prove beneficial for future research in this area. Additionally, reporting adverse effects, even with zero occurrence, should become more common for VR research, as it is unclear whether they appear only very rarely or they simply go unreported.

The results of this review predominantly focus on short-term change, with longer-term effects unknown. With only three studies employing follow-up measurements, the retention of positive changes beyond immediately after the intervention needs further research in order to establish the effectiveness of VR for our domains.

Since self-compassion and self-protection were identified as the antidotes to self-criticism within the latest research findings of Emotion-focused Therapy [20], we propose developing a novel VR intervention to decrease levels of self-criticism through evoking self-compassion and self-protection based on the two-chair technique [73,74].

Consistent with the wider affordability of VR as of 2016 [3], the number of studies looking at self-compassion and self-criticism has significantly grown this year. However, this is not the case for self-protection/assertiveness, as the latest study in this domain was published in 2014. This highlights a large research gap and an imminent need to expand on this area, especially if published studies suggest an existing potential. In fact, considering the low overall number of studies in this review, this statement can be applied to all three domains. More studies, especially randomized controlled trials (there were only two in this review), are needed to investigate the effect of VR on self-criticism, self-compassion and self-protection. Future trials should most importantly include larger samples, both clinical and non-clinical, and look at longer-term effects through follow-up periods; additionally, VR interventions need to be replicable and clearly embedded in theory to stimulate wider research; and finally, since age (e.g., [75]) and gender (e.g., [76]) may play a part in the way VR experience is processed, these should be investigated too.

## 5. Limitations

This systematic review has been conducted by only two reviewers. There is always a possibility that some studies may have been missed during the process. Limited published literature limits broader conclusions. It is difficult to estimate how many studies containing non-significant results may have gone unpublished or were published in different languages except for English. Not all studies reported all details, making a sound judgment difficult, and not all of the contacted authors responded to our questions to specify their research study. The selected studies varied greatly, leaving the final synthesis open to doubt. Finally, this study was not pre-registered. For future systematic reviews looking at this research area, we suggest more reviewers be involved and pre-register the protocol in order to mitigate any additional bias that may have arisen from these limitations.

## 6. Conclusions

A systematic review, guided by the PRISMA 2020 principles [36], has been conducted in order to provide an overview of the effectiveness of VR interventions aiming to increase self-compassion and self-protection and decrease self-criticism. In all three domains, positive results have been identified, supporting the notion that VR will become more established in mental health practice in the near future. However, any generalizations have to be inferred very carefully because the number of studies identified in this review is limited and they vary in design and application. Larger sample randomized controlled trials with a longer term follow-up period are required in order to establish robust evidence for VR interventions for all three outcome measures.

## Figures and Tables

**Figure 1 ijerph-20-02592-f001:**
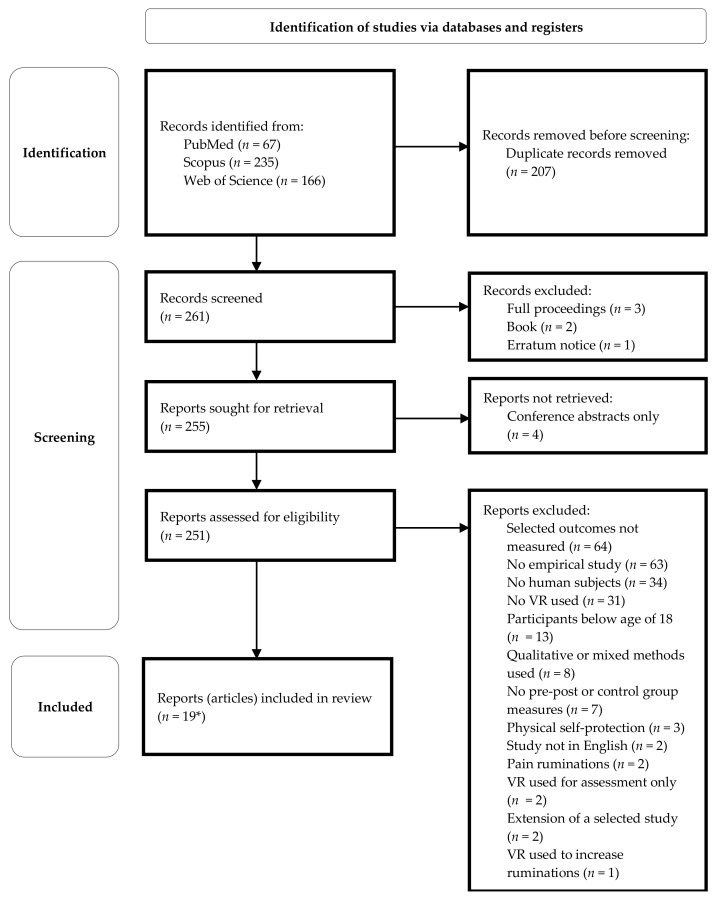
PRISMA 2020 flow diagram of selected reports. * Two articles on the final list had been identified as reports of the same research; therefore, the final number of selected reports is 18.

**Table 1 ijerph-20-02592-t001:** Breakdown of search records according to topic and databases applied.

	Virtual Reality + Self-Compassion	Virtual Reality + Self-Criticism	Virtual Reality + Self-Protection	Σ
PubMed	28	16	23	67
Scopus	90	46	99	235
Web of Science	79	34	53	166
Σ	197	96	175	468

All data is shown before duplicates were removed.

**Table 2 ijerph-20-02592-t002:** Final summary of selected studies.

Reference	Study Type (Defined According to the Purpose of This Systematic Review)	Intervention (Length in Minutes)/Overall VR Exposure in Minutes	VR Exposure in Control Group(s)	VR Intervention Description	VR Type	Sample (age)	Intervention/Control Groups Size	Selected Outcomes (Measure)	Results of Outcomes
Ascone et al., 2020 [45]	Randomized, longitudinal (between-groups) study	Psycho-education + VR exposure with embodiment (10)/10	Yes	Mission to the moon to explore mysterious interactive nebula	Immersive (HMD)/interactive	Students with mildly elevated paranoia symptoms (18+)	12/9	Self-compassion (brief, state-adapted SCS based on [46])	Intervention and control groups analyzed independently (within-group). Significant positive change in self-compassion post-psycho-education and VR exposure in intervention group (*χ^2^* = 16.93, *p* < 0.001). No significant change in self-compassion in control group (*χ^2^* = 3.16, *p* = 0.206). No effect size reported.
Cebolla et al., 2019 [42] (7A) (T2,T3)	Randomized, longitudinal (between-groups) study	Virtual body swap/embodiment I. (5) + Self-compassion audio meditation (15) + Virtual body swap/embodiment II. (5–7)/10–12	No	I. Being in someone else’s body; II. Hugging oneself from third-person perspective	Immersive (HMD)/interactive	University students (18+)	8/8	Compassion toward self (positive qualities towards self) (SOFI)	Increase in positive qualities towards self; however, no significant interaction between intervention and control groups, [*F*(1, 14) = 0.66, *p* = 0.429, *η^2^p* = 0.05]. Large within-group effect size for positive qualities towards self in intervention group (*d* [95% CI] = −0.82); CI for control group included 0.
Cebolla et al., 2019 [42] (7B) (T1,T4)	Randomized, longitudinal (between-groups) study	Virtual body swap/embodiment (5) + Self-compassion audio meditation (15) + Virtual body swap/embodiment (5–7) + 2 weeks of voluntary self-compassion audio meditation/10–12	No	I. Being in someone else’s body; II. Hugging oneself from third-person perspective	Immersive (HMD)/interactive	University students (18+)	8/8	Self-compassion as part of self-care behaviors (MSCS)	Increase in self-compassion with no significant interaction effects between intervention and control groups for Self-compassion and Purpose subscale [*F*(1, 14) = 1.34, *p* = 0.266, *η^2^p* = 0.09]. Large within-group effect size for this subscale in the intervention group (*d* [95% CI] = −0.77); CI for control group included 0.
Falconer et al., 2014 [35]	Longitudinal (between-groups) study; information on group randomization not reported	Psycho-education + Embodiment as an adult (3) + VR scenario I. + Embodiment as a child (3) + VR scenario II./not all durations reported	Yes	I. Speaking to a distressed virtual child; II. Receiving one’s own soothing message from the child’s perspective	Immersive (HMD)/interactive	Female undergraduate students with higher levels of self-criticism (18+)	22/21	State self-compassion (SCCS), state self-criticism (SCCS)	Self-compassion increased in 1PP group; significant main effect of time [*F*(1, 41) = 14.33, *p* < 0.001, *η^2^p* = 0.26], with significant interaction between 1PP and 3PP groups [*F*(1, 41) = 15.87, *p* < 0.001, *η^2^p* = 0.28]. Self-criticism decreased in both 1PP and 3PP groups; main effect of time [*F*(1, 41) = 42.1, *p* < 0.001, *η^2^p* = 0.51], with no significant interaction between 1PP and 3PP groups [*F*(1, 41) = 3.46, *p* = 0.07, *η^2^p* = 0.8].
Falconer et al., 2016 [8]	Longitudinal (within-group) study with 4-week follow-up	Psycho-education + 3× once weekly [Embodiment as an adult (2) + VR scenario I. (2) + Embodiment as a child (2) + VR scenario II. (2)]/24	N/A	I. Speaking to a distressed virtual child; II. Receiving one’s own soothing message from the child’s perspective	Immersive (HMD)/interactive	Current major depressive disorder (18+)	15/-	Self-compassion, self-criticism (SCCS)	Significant linear increase in self-compassion [*F*(1, 12) = 6.65, *p* < 0.02, *η^2^p* = 0.36] and significant linear decrease in self-criticism [*F*(1, 12) = 23.41, *p* < 0.001, *η^2^p* = 0.66] between pre, post, and 4-week follow-up.
Hur et al., 2021 [47]	Longitudinal (within-group) study	6× [Introduction & VR meditation (5) + VR exposure (7–8) + VR meditation & psycho-education (3)]/90	No	Participants introduce themselves in student group meeting. Scenarios vary in difficulty.	Immersive (HMD)/interactive	Social anxiety disorder (18+)	21/21 (control group not relevant for selected variable)	Negative post-event ruminations (PERS)	Negative post-event ruminations significantly decreased between pre- and post-measures (*Z* = −3.32, *p* < 0.001, *r* = 0.51).
Kim et al., 2020 [48]	Longitudinal (within-group) study	6× [Introduction & VR meditation + VR exposure + psycho-education]/90	No	Participants introduce themselves in student group meeting. Scenarios vary in difficulty.	Immersive (HMD)/interactive	Social anxiety disorder (18+)	32/33 (control group not relevant for selected variable)	Negative post-event ruminations (PERS)	Negative post-event ruminations significantly decreased after the intervention [*F*(2.730) = 6.97, *p* < 0.001]. Effect size not reported.
Klinger et al., 2004; 2005 [41,40]	Non-randomized, longitudinal (between-groups) study	Introduction + 11× [VR assessment and/or VR exposure (20)]/220	No	Four different social situations causing anxiety.	Non-immersive (PC)/interactive	Outpatients with social phobia (18+)	18/18	Assertiveness (RAS, SCIA)	Assertiveness significantly increased in both VR and group CBT conditions; *F*(1, 34) = 36.30, *p* < 0.001 for RAS and *F*(1, 34) = 65.77, *p* < 0.001 for SCIA/Assertiveness subscale; CBT shows greater improvement for both subscales. No significant interaction effects were found; *F*(1, 34) = 2.66, *p* > 0.05, *η^2^* = 0.07 for RAS, and *F*(1, 34) = 0.81, *p* > 0.05, *η^2^* = 0.02 for SCIA/Assertiveness subscale.
Modrego-Alarcón et al., 2021 [49]	Randomized controlled trial with 6-month follow-up	6× once weekly [Mindfulness meditation (75) + VR meditation (app.7.5)]/46	No	Observing objects (tree, leaves, lemon, etc.), human figure, walking through a landscape, taking part in a university exam.	Immersive (HMD)/non-interactive	University students (18+)	70/65/53	Self-compassion (SCS)	Self-compassion significantly increased from pre-to-post (*B* = 7.93, *p* < 0.001, *d* = 0.94) and from pre- to 6-month follow-up (*B* = 12.42, *p* < 0.001, *d* = 1.47) in VR mindfulness group, compared with relaxation. No significant interaction effect between VR and non-VR mindfulness group; pre-to-post (*B* = 0.86, *p* = 0.71, *d* = 0.1) and pre- to 6-month follow-up (*B* = −3.11, *p* = 0.19, *d* = −0.36).
Navarrete et al., 2021 [50] (T2,T3)	Randomized, longitudinal (between-groups) study	Embodiment (5) + Embodiment with audio (4.5) + Audio (3)/9.5	No	Embodying a patient with panic attack disorder and directing compassion towards him.	Immersive (HMD)/interactive	Healthcare students and professionals (18+)	21/20	State self-compassion (VAS-SC, 2 questions), state self-criticism (VAS-SC, one question)	Self-compassion decreased a little in intervention group [*t*(20) = 0.21, *p* = 0.838, *η^2^* = 0.00], no significant interaction with control group [*F*(1, 38) = 0.35, *p* = 0.556, *η^2^* = 0.00]. Self-criticism decreased in intervention group [*t*(19) = 1.85, *p* = 0.079, *η^2^* = 0.15], no significant interaction with control group [*F*(1, 37) = 0.03, *p* = 0.868, *η^2^* = 0.00].
O’Gara et al., 2022 [51]	Longitudinal (within-group) study	3× [VR exposure (10) at least a week apart]/30	N/A	360° video of a beach, animated mountain, or animated forest scene, with choice of a male or female guiding voice delivering breathing and CMT exercises	Immersive (HMD)/non-interactive	Cancer patients (18+)	10/-	Self-compassion (SCS)	No significant changes between baseline, VR1, VR2, and VR3 sessions for any SCS subscales (SK: *χ*^2^(3) = 0.733, *p* = 0.866; SJ: *χ*^2^(3) = 2.133, *p* = 0.545; CH: *χ*^2^(3) = 0.976, *p* = 0.807; IS: *χ*^2^(3) = 2.018, *p* = 0.569; MF: *χ*^2^(3) = 5.23, *p* = 0.156; OI: *χ*^2^(3) = 4.417, *p* = 0.22). No significant changes between baseline and VR3 sessions for any SCS subscale (SK: *Z* = −1.011, *p* = 0.312; SJ: *Z* = −0.978, *p* = 0.328; CH: *Z* = −0.224, *p* = 0.823; IS: *Z* = −1.261, *p* = 0.207; MF: *Z* = −1.605, *p* = 0.108; OI: *Z* = −1.43, *p* = 0.153). Total SCS score and effect size not reported.
Park et al., 2011 [52]	Randomized controlled trial	10× [3× VR role plays, including modeling by therapist and positive or corrective feedback (90)]/duration details not reported	No	Conversation, assertiveness, and emotional expression skills trained in common social situations.	Immersive (HMD)/interactive	Inpatients with schizophrenia (18+)	32/31	Assertiveness (RAS)	Improvements in assertiveness in both VR and non-VR social skills training group; significant time effect, *F*(1, 62) = 26.17, *p* < 0.001, *η^2^p* = 0.3; and significant time x group interaction, *F*(1, 62) = 4.96, *p* = 0.03, *η^2^p* = 0.07. VR group shows greater improvement on RAS score.
Park & Ogle, 2021 [53]	Longitudinal (within-groups) study	4× [Body positivity program (120)] + Virtual avatar experience (4–13)/4–13	No	Presenting anthropometrically accurate avatars of participants themselves in four different contextual backgrounds.	Non-immersive (PC)/interactive	Female undergraduate students with body image concerns (18+)	9/9 (control group not relevant for selected variable)	Self-compassion (SCS-SF)	Purpose of control group unclear. Repeated-measures ANOVA (most likely for experimental group data) shows significant improvements in self-compassion between baseline and post-VR (*p* = 0.000) and between pre-VR and post-VR (*p* = 0.041). No inferential statistics or effect size reported.
Price & Anderson, 2011 [54]	Randomized, longitudinal (between-groups) study	8× [VR exposure]/not reported	No	Challenging public speaking situations, such as classroom or auditorium.	Immersive (HMD)/interactive	Individuals diagnosed with social anxiety disorder (18+)	32/33/25	Ruminations (RQ)	VR exposure (*β* = −8.82, *p* < 0.01) and group CBT (*β* = −9.85, *p* < 0.01) both significantly better than waiting list controls. VR and CBT together had a large effect size compared with waiting list (33% of the variance at post-intervention). After controlling for pre-intervention scores, no significant difference between group CBT and VR at post-intervention (*β* = 0.62, *p* = 0.62).
Riva et al., 2001 [11]	Randomized, longitudinal (between-groups) study	7× once weekly [VR exposure (50)] + low-calorie diet + physical training (minimum twice 30 min walk a week)/350	No	Exposure to environments, potentially eliciting abnormal eating behaviors.	Immersive (HMD)/interactive	Female patients with overweight issues (18+)	28 participants in total (estimated as 14/14)	Assertiveness (AI)	Significant improvements in ability to engage in anxiety-provoking behaviors (*p* = 0.014) within intervention group, and significant improvements in ability to engage in anxiety-provoking behaviors in intervention group compared with control group (*p* = 0.000). No other inferential statistics or effect size reported.
Riva et al., 2002 [55]	Randomized, longitudinal (between-groups) study	7× once weekly [VR exposure (50)] + low-calorie diet + physical training (minimum twice 30 min walk a week)/350	No	Exposure to environments potentially eliciting abnormal eating behaviors.	Immersive (HMD)/interactive	Female patients with binge eating disorder (18+)	20 participants in total (estimated as 10/10)	Assertiveness (AI)	Significant improvements in ability to engage in anxiety-provoking behaviors (*p* = 0.038) within intervention group and non-significant improvements in ability to engage in anxiety-provoking behaviors in intervention group compared with control group (*p* = 0.063). No other inferential statistics and effect size reported.
Roy et al., 2003 [56]	Non-randomized, longitudinal (between-groups) study	Introduction + 11× [VR assessment and/or VR exposure (20)]/220	No	Four different social situations causing anxiety.	Non-immersive (PC)/interactive	Social phobia (18+)	4/6	Assertiveness (RAS)	Based on descriptive statistics, assertiveness increased in both VR and group CBT conditions. No inferential statistics, significance levels, or effect size reported.
Rus-Calafell et al., 2012 [57]	Case study	16× [Content introduction (30) + VR exposure (30)]/480	N/A	Exposure to common daily situations, such as going to a shop or dealing with an angry security guard.	Non-immersive (PC + 3D glasses + headphones)/interactive	Schizophrenia outpatient (18+)	1/-	Assertiveness (AI, number of assertive behaviors observed)	Improvements in ability to engage in anxiety-provoking behaviors pre- and post-intervention (no inferential statistics reported). Significant increase in assertive behaviors pre- to post-intervention (*Z* = −3.28, *p* < 0.05). No effect size reported.
Rus-Calafell et al., 2014 [15]	Longitudinal (within-group) study with 4-month follow-up	16× [Content introduction (30) + VR exposure (30)]/480	N/A	Exposure to common daily situations, such as going to a shop or dealing with an angry security guard.	Non- immersive (PC + 3D glasses + headphones)/interactive	Schizophrenia/schizoaffective disorder outpatients (18+)	12/-	Assertiveness (AI, number of assertive behaviors observed)	Significant improvements in ability to engage in anxiety-provoking behaviors pre- and post- a 4-month follow-up, with large effect [*F*(2, 22) = 70.79, *p* < 0.01, *d* = 0.87]. Same results observed for number of assertive behaviors observed [*F* (3, 33) = 139.76, *p* < 0.01]; no effect size reported.

AI: Assertion Inventory [58]; CMT: Compassionate Mind Training [59]); HMD: Head-Mounted Display; MSCS: Mindfulness Self-Care Scale [43]; PERS: Post-Event Rumination Scale [60,61,62]; RAS: Rathus Assertiveness Schedule [63]; RQ: Rumination Questionnaire [64]; SCIA: Social Contexts Inducing Anxieties [65]; SCCS: Self-compassion and Self-criticism Scales [66]; SCS: Self-Compassion Scale [67]; SCS-SF: Self-compassion Scale Short Form [68]; SOFI: Self-Other Four Immeasurable Scale [44]; T1,T2,T3,T4: Time data collection points; VAS-SC: Visual Analogue Scales for State Changes [69]; VR: Virtual Reality.

**Table 3 ijerph-20-02592-t003:** Risk of bias summary for the selected studies.

	Random Sequence Generation/Allocation Concealment(Selection Bias)	Blinding of Participants and Personnel (Performance Bias)	Blinding of Outcome Assessment (Detection Bias)	Incomplete Outcome Data Addressed (Attrition Bias)	Selective Reporting (Reporting Bias)	Other Biases
Ascone et al. (2020) [45]	U	H	H	L	U	H
Cebolla et al., 2019 [42] (7A) (T2,T3)	U	H	H	L	U	H
Cebolla et al., 2019 [42] (7B) (T1,T4)	U	H	H	L	U	H
Falconer et al., 2014 [35]	U	H	H	U	U	H
Falconer et al., 2016 [8]	H	H	H	L	U	H
Hur et al., 2021 [47]	H	H	H	H	U	H
Kim et al., 2020 [48]	H	H	H	H	U	H
Klinger et al., 2004; 2005 [41,40]	H	H	H	L	U	H
Modrego-Alarcón et al., 2021 [49]	L	H	H	U	L	H
Navarrete et al., 2021 [50] (T2,T3)	U	H	H	U	U	H
O’Gara et al., 2022 [51]	H	H	H	H	U	H
Park & Ogle, 2021 [53]	U	H	H	L	U	H
Park et al., 2011 [52]	U	H	H	H	U	H
Price & Anderson, 2011 [54]	U	H	H	H	U	H
Riva et al., 2001 [11]	U	H	H	L	U	H
Riva et al., 2002 [55]	U	H	H	L	U	H
Roy et al., 2003 [56]	H	H	H	L	U	H
Rus-Calafell et al., 2012 [57]	H	H	L	N/A	U	H
Rus-Calafell et al., 2014 [15]	H	H	L	H	U	H

Selection bias was coded as unclear (U) if both random sequence generation and concealment were coded as having unclear risk or if one was coded as unclear and the other as low risk (L). Other bias was coded as high (H) if at least one other bias was identified as high risk (see Appendix B for details).

**Table 4 ijerph-20-02592-t004:** GRADE certainty assessment for measured outcomes.

Outcome	Baseline Assessment	Risk of Bias	Inconsistency	Indirectness	Imprecision	Publication Bias	Overall Assessment
Self-criticism/Ruminations	LOW—no randomized controlled trials for this outcome	Downgrading by one point—out of six identified bias categories, all studies show between three and five high risks of bias and one to three unclear risks of bias	No downgrading out of six—five studies report significant results, one study reports non-significant results; effect in all studies points in the same direction	Downgrading by one point—some variability in content and duration of intervention; three out of six studies had no control group for this variable	No downgrading—total number of participants is 232 (<400); however, five studies out of six report large effect size, and one study	No downgrading—results come predominantly from smaller studies; however, this is an emerging area of research, with most studies published since 2016; publication bias possible but not strongly suspected	VERY LOW
Self-compassion/Positive qualities towards self	LOW—only one study out of nine was a randomized controlled trial	Downgrading by one point—out of six identified bias categories, all studies show between three and five high risks of bias and one to three unclear risks of bias	No downgrading—out of nine studies, five report significant results, four report non-significant results; effect in seven studies points in the same direction, one non-significant study points in opposite direction and another non-significant study does not provide information on direction	Downgrading by one point—variability in content, number of interventions, length of exposure, and employed technology; in two out of nine studies, VR did not constitute central part of the intervention	Downgrading by one point—total number of participants is 368 (<400); three studies out of nine did not report effect size, thereby providing insufficient information about effect sizes reported across the studies	No downgrading—results come predominantly from smaller studies; however, this is an emerging area of research, with most studies published since 2016; publication bias possible but not strongly suspected	VERY LOW
Self-protection/Assertiveness	LOW—only one study out of seven was a randomized controlled trial	Downgrading by one point—out of six identified bias categories, all studies show between three and four high risks of bias and one to two unclear risks of bias	No downgrading—out of seven, five studies show significant results, one study does not report significance fully, and one study does not report significance at all; effect in all studies points in the same direction	Downgrading by one point—variability in content, number of interventions, length of exposure and employed technology	Downgrading by one point—total number of participants is 170 (<400); five out of seven studies did not report any effect size, thereby providing insufficient information about effect sizes reported across the studies	Downgrading by one point—results come mainly from smaller studies; none of the studies had been preregistered; only one study reports sample size power calculation; no studies published since 2014; publication bias suspected	VERY LOW

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
