# Peer review of "The Effects of Virtual Reality on Enhancement of Self-Compassion and Self-Protection, and Reduction of Self-Criticism: A Systematic Review"

_ijerph, 2023, doi:10.3390/ijerph20032592_

Round 1
Reviewer 1 Report (Previous Reviewer 1)
Dear authors,
In the attached document I have replied to your comments (in green).
Regards,

Author Response
Dear reviewer,
Thank you for personally reviewing our manuscript and for giving us the opportunity to respond to your comments.
We have made the relevant changes as detailed below and would like to submit our amended manuscript for submission in International Journal of Environmental Research and Public Health. These changes have been highlighted in the manuscript through track changes. We hope you find our response detailed and satisfactory. If you have any further questions or comments, please do not hesitate to contact me.
Best wishes
authors
- was the review registered in any registry of systematic reviews?
Our response: This section has now been added to the limitations section: 'Finally, this study was not pre-registered. For future systematic reviews looking at this research area, we suggest more reviewers to be involved and to pre-register the protocol in order to mitigate any additional bias that may have arisen from these limitations.'
- In relation to the type of VR, it would be advisable to specify what type of activities were selected: physical, cognitive, specific treatment activities with clinical VR...?
Our response: We are unsure whether we understand this point correctly. In Table 2, column 5, we explain the content of scenarios applied in each of the selected studies. These scenarios are either passive (non-interactive) or require active involvement of study participant (interactive). These scenarios are psychological interventions; there is no physical aspect/performance involved.
- No studies reported adverse effects?
Our response: We added the following paragraph to the results section: ' Adverse effects. In addition to the analysis above, we also screened the studies of this review for any reports of adverse side effects as a result of taking part in VR, also known as cybersickness (Kennedy & Stanney, 1996). One study (O'Gara et al., 2022) reported adverse side effects (nausea and dizziness) in two participants, however this may not have been the result of VR intervention as they were also attending chemotherapy at the same time. Three studies (Park et al., 2011; Riva et al., 2001; Riva et al., 2002) reported no cybersickness in any of their participants. The remaining studies did not address VR-related adverse side effect in their studies.'
Also, we added the following sentence to the discussion: ' Additionally, reporting adverse effects, even with zero occurrence, should become more common for VR research, as it is unclear whether they appear only very rarely, or they simply go underreported.'
- age and gender of participants?
Our response: In terms of gender, only four studies (one measuring self-criticism and self-compassion, one measuring self-compassion and two measuring assertiveness) were conducted with female only participants and we state this information in Table 2, column 7. The rest of the studies contains various ratios of men and women. It is very difficult to make any assumptions from this.
In terms of age, seven studies report age in the format of mean and standard deviation only, making it impossible to be sure that no participant is over 70; two studies have an age ceiling of 65 as their inclusion criterion; eight studies report age range with none of the oldest participant being over 61; and one study reports and age range with the oldest participant being 77. As in the case of age, it is very difficult to make any assumptions from this.
However, we added the following sentence to the discussion: '... and finally, since age (e.g., Liu et al., 2020) and gender (e.g., Felnhofer et al., 2012) may play part in the way VR experience is processed, these should be investigated too.'
- I cannot find information on the characteristics of the intervention carried out in each study: number of sessions, time of each session, subsequent follow-up
Our response: This information is in Table 2, column 2 and 3
- what do you suggest for the design of RCTs that aim to use VR to improve the factors you have studied?
We've added the following summary to the discussion: ' Future trials should most importantly include larger samples, both clinical and non-clinical and look at longer-term effects through follow-up periods; additionally, VR interventions need to be replicable and clearly embedded in theory to stimulate wider research; and finally, since age (e.g., Liu et al., 2020) and gender (e.g., Felnhofer et al., 2012) may play part in the way VR experience is processed, these should be investigated too.'

Reviewer 2 Report (Previous Reviewer 3)
The new version of the manuscript was revised according to my previous comments. Hence, I recommend the paper to be accepted in its current version.
Author Response
Thank you for personally reviewing our manuscript. Best wishes authors
This manuscript is a resubmission of an earlier submission. The following is a list of the peer review reports and author responses from that submission.
Round 1
Reviewer 1 Report
Dear authors,
I find your work very interesting. Virtual reality is a technique of increasing use and impact in treatment in the field of mental health. I congratulate you for the choice of the topic of the systematic review.
I would like to make some suggestions about your work:
1. Introduction.
The authors give an adequate introduction to the concepts of Virtual Reality, depression, self-criticism and self-pity.
However, the introduction does not mention how VR is used to improve self-criticism, self-compassion and self-protection. There are questions that the introduction to a systematic review needs to answer: what studies have been conducted, what results have been obtained, do these results disagree? And most importantly: why is such a systematic review necessary? The authors do not answer these questions in their introduction.
2. Methodology:
- was the review registered in any registry of systematic reviews?
- Could they elaborate more on the VR technology they have selected in their studies? Immersive, non-immersive? This point is important
- The date the review ended is old... it is not clear to me if it should be updated Chapter IV: Updating a Review (Cochrane) indicates that SRs should be reviewed after 6 months. Established the authors some system of alarm that inform them of new works published until the moment of the sending of the R sauna magazine?
3. Results
It would be useful to add the age and gender of the participants.
4. Discussion
No studies reported adverse effects?
What do the authors suggest for future studies? Perhaps suggestions regarding sample size, follow-up, type of VR technology, number of sessions, exposure time? .... This information should always be present in the discussion of a systematic review. It is the information researchers are looking for.
Best regards,
Author Response
Dear editor and reviewers,
We would like to thank you very much for your time and effort spent reviewing our systematic review. Please find below our comments to your suggestions:
Reviewer 1
What studies have been conducted, what results have been obtained, do these results disagree?
Our response: We believe this to be the first systematic review focusing on these concepts applied in virtual reality. This is the reason why no previous studies have been mentioned in the introduction. We state this at the very end of the introduction.
Why is such a systematic review necessary?
Our response: This is the first systematic review focusing on self-compassion, self-criticism and self-protection applied in virtual reality. As results of using virtual reality for mental health interventions are very promising, we were particularly interested how effective VR is for these three concepts.
Was the review registered in any registry of systematic reviews?
Our response: No, it hasn't been pre-registered. We state this at the very end of our abstract.
Could they elaborate more on the VR technology they have selected in their studies? Immersive, non-immersive?
Our response: We define only immersive/non-immersive and interactive/non-interactive VR technology as, likely, this is the most straightforward way to understand VR technology from a non-technical user-experience point of view. There are, of course, other parameters (e.g., display resolution, manufacturer, etc.) that could be looked at when classifying VR technology, however this would perhaps make a lot more sense once there are more studies published.
The date the review ended is old... it is not clear to me if it should be updated Chapter IV: Updating a Review (Cochrane) indicates that SRs should be reviewed after 6 months. Established the authors some system of alarm that inform them of new works published until the moment of the sending of the R sauna magazine?
Our response: It takes very long time to conduct a systematic review. We are only two researchers with limited resources and updating the systematic review would mean more delays as it would have a knock-on effect on the whole article every time, we update it.
It would be useful to add the age and gender of the participants.
Our response: We selected studies with adult participants only so all studies have participants aged 18 or above. This is stated in Table 2 in the 'Sample (age)' column. We are unsure if adding gender would provide any additional insight - we had the EFT framework in mind when we conducted this review where gender plays no part.
No studies reported adverse effects?
Our response: No, we haven't come across any adverse VR treatment effects with regards to our outcomes in selected studies. However, few studies, such as Kim et al. (2020), mention increased clinical symptomatology as part of their intervention, arguing that it is often necessary for any mental health treatment to exacerbate symptoms for it to have therapeutic effect. This would, therefore, not apply specifically to VR.
What do the authors suggest for future studies? Perhaps suggestions regarding sample size, follow-up, type of VR technology, number of sessions, exposure time?
Our response: We do suggest randomised controlled trials to establish solid evidence base. For each of the outcomes, we summarised respective variables, but it would be very hard to make any specific suggestions (besides those we made) as there isn't enough evidence to base them on.
Reviewer 2 Report
The authors conducted a systematic review on VR interventions towards reducing self-criticism and/or increasing self-compassion and self-protection. The review was conducted rigorously following established guidelines (PRISMA, Cochrane Collaboration’s tool, GRADE, etc). Overall, congratulations to the authors for producing a well-written manuscript!
There are only minor comments and suggestions.
With regards to the writing, sometimes “VR” is used and sometimes it’s “virtual reality”. Suggest to keep to VR throughout since the abbreviation was introduced at the start
The discussion in its current form could be improved with greater engagement with the literature on self-compassion/protection/criticism. While the authors have done a good job summarizing the findings, the theoretical contributions could be strengthened. For example, do the findings highlight certain strengths of VR over traditional interventions? It would be good to see a deeper discussion on how VR compares to traditional interventions. In addition, how do the findings extend theories in the field (e.g., emotion focused therapy)? The authors could also develop testable hypotheses and suggest key areas for future research.
Author Response
Dear editor and reviewers,
We would like to thank you very much for your time and effort spent reviewing our systematic review. Please find below our comments to your suggestions:
Reviewer 2
With regards to the writing, sometimes “VR” is used and sometimes it’s “virtual reality”. Suggest to keep to VR throughout since the abbreviation was introduced at the start
Our response: This has now been amended throughout the article.
Do the findings highlight certain strengths of VR over traditional interventions? It would be good to see a deeper discussion on how VR compares to traditional interventions.
Our response: Only three studies used traditional intervention (CBT) as a control group which is a very low number of studies to make any conclusions from. Moreover, no significant difference was found between VR-based and traditional CBT treatments in these studies.
How do the findings extend theories in the field (e.g., emotion focused therapy)? The authors could also develop testable hypotheses and suggest key areas for future research.
Our response: Thank you, we added a few sentences about future research based on EFT.
Reviewer 3 Report
The paper presents a systematic review of the studies testing VR interventions aimed at reducing self-criticism and enhancing self-compassion and self-protection. The approach and the methods used in the selection of the studies included in the review are correct, and the indicators / variables extracted from the analyses of these studies are comprehensive and generally well discussed. Therefore, I believe the paper would represent a solid contribution to the extant knowledge on the benefits of VR-based interventions targeting psychological factors in clinical and non-clinical settings. There are only a few changes that I believe would enhance the strength of argumentation of the manuscript and its comprehensiveness.
The argument in the Introduction about the benefits of self-compassion and self-protection for diminishing self-criticism and consequently depression and anxiety should complemented with empirical results that attest to these effects, besides the theoretical propositions that are reviewed in the current version of the manuscript.
Moreover, whether the studies selected in the systematic review also include or measure depression or anxiety, beyond the three target dimensions (self-compassion, criticism, protection), should also be commented in the Results and/or Discussion section, in line with the main conceptual framework of the paper in which these represent mediators of the effects of VR exposure on depression / anxiety.
Line 145 the use of these specific exclusion criteria should be justified. Moreover, why was only “compassion directed towards animals” used as exclusion criterion and not compassion towards other targets than oneself, since the focus of the review is on self-compassion?
Line 191 “To assess attrition, we followed the five-and-20 rule of thumb” – this should be briefly explained.
Line 195 “Certainty assessment was completed using the GRADE framework” – this should be briefly explained.
To ensure the consistency of reporting, the types / content of scenarios used should also be commented for self-protection (section 7.2)
Some general conclusion or at least comment about the differences in effects / efficiency across the three dependent variables between immersive and non-immersive VR interventions, respectively among the various types of scenarios used should be included in the Discussion.
Author Response
Dear editor and reviewers,
We would like to thank you very much for your time and effort spent reviewing our systematic review. Please find below our comments to your suggestions:
Reviewer 3
The argument in the Introduction about the benefits of self-compassion and self-protection for diminishing self-criticism and consequently depression and anxiety should complemented with empirical results that attest to these effects, besides the theoretical propositions that are reviewed in the current version of the manuscript.
Our response: Thanks a lot, we added information about the previous research studies on effectiveness of EFT in diminishing self-criticism via self-compassion and self-protection.
Moreover, whether the studies selected in the systematic review also include or measure depression or anxiety, beyond the three target dimensions (self-compassion, criticism, protection), should also be commented in the Results and/or Discussion section, in line with the main conceptual framework of the paper in which these represent mediators of the effects of VR exposure on depression / anxiety.
Our response: Thank you for your suggestion. Targeting also depression and anxiety would yield very different results which will not be compatible with the goal of our article. We solely focused on self-criticism, self-protection, and self-compassion as these three concepts create the base of the change model in Emotion-focused therapy. Rarely, systematic reviews are aimed on too many concepts, as it is already very exhausting process to do.
Line 145 the use of these specific exclusion criteria should be justified. Moreover, why was only “compassion directed towards animals” used as exclusion criterion and not compassion towards other targets than oneself, since the focus of the review is on self-compassion?
Our response: Coming from the psychotherapeutic point of view, we focused solely on compassion related to people rather than other entities. Although not yet thoroughly evidenced, compassion and self-compassion are likely very similar processes, only directed different ways - either in or out. This is why we didn't want to exclude compassion.
Line 191 “To assess attrition, we followed the five-and-20 rule of thumb” – this should be briefly explained.
Our response: This has now been added to the article.
Line 195 “Certainty assessment was completed using the GRADE framework” – this should be briefly explained.
Our response: This has now been added to the article.
To ensure the consistency of reporting, the types / content of scenarios used should also be commented for self-protection (section 7.2)
Our response: We do summarise this in the section 3.3 by saying 'Scenarios included daily social situations and exposure to challenging stimuli.'
Some general conclusion or at least comment about the differences in effects / efficiency across the three dependent variables between immersive and non-immersive VR interventions, respectively among the various types of scenarios used should be included in the Discussion.
Our response: We have now added a 'Conclusions' section to the article.
Thank you again for your comments and we are looking forward to your feedback.
Round 2
Reviewer 1 Report
Dear authors,
I have not found any changes in relation to the proposals made in the first revision.
Regards,
Reviewer 3 Report
I thank the authors for their effort in revising the manuscript. Still, there are two issues left unresolved:
1. On of my previous comments was “why was only “compassion directed towards animals” used as exclusion criterion and not compassion towards other targets than oneself, since the focus of the review is on self-compassion?”. The authors response states” Coming from the psychotherapeutic point of view, we focused solely on compassion related to people rather than other entities. Although not yet thoroughly evidenced, compassion and self-compassion are likely very similar processes, only directed different ways - either in or out.”. This is outside the focus of the study, as the authors state on line 112 that “This systematic review summarises existing scientific literature on VR interventions applied to decrease self-criticism or increase self-compassion or self-protection,” – self-compassion, not compassion in general. Coming from a psychological point of view, the presumed effects of EFT are due not to the process of compassion itself, but on its change in self-directed emotions and appraisals (among others, by alleviating self-criticism). This is in line with authors’ statements in the paper: “As the antidote to the pain that self-criticism (or negative self-treatment) causes, two adaptive processes were identified—self-compassion and self-protection. Self-compassion is best understood as an inward directed compassion (Neff & Tirch, 2013).” By this reasoning, self-compassion specifically and not compassion in general should be the focus of the studies included in the review. If the authors propose (as they already did in the actual review process) the extension of the scope of the review to include compassion towards other people (which conflicts with their own current description of the aim of the study), this should be substantially justified. Alternatively, the studies and / or results that are only pertinent to compassion towards others should be eliminated from the review.
2. One of my previous comments stated that “Some general conclusion or at least comment about the differences in effects / efficiency across the three dependent variables between immersive and non-immersive VR interventions, respectively among the various types of scenarios used should be included in the Discussion.”. The authors’ reply was “We have now added a 'Conclusions' section to the article.” That is true, but the newly added text is not relevant to the issues I raised in my comment. Therefore, this point still remains to be addressed.